# The Potential of Phenothiazines against Endodontic Pathogens: A Focus on *Enterococcus-Candida* Dual-Species Biofilm

**DOI:** 10.3390/antibiotics11111562

**Published:** 2022-11-05

**Authors:** Nicole de Mello Fiallos, Ana Luiza Ribeiro Aguiar, Bruno Nascimento da Silva, Mariana Lara Mendes Pergentino, Marcos Fábio Gadelha Rocha, José Júlio Costa Sidrim, Débora Castelo Branco de Souza Collares Maia, Rossana de Aguiar Cordeiro

**Affiliations:** 1Department of Pathology and Legal Medicine, Federal University of Ceará, Fortaleza 60430-160, Brazil; 2Department of Veterinary Sciences, School of Veterinary, State University of Ceará, Fortaleza 60714-903, Brazil

**Keywords:** *Enterococcus faecalis*, *Candida albicans*, biofilm, phenothiazines, periapical periodontitis

## Abstract

Persistent apical periodontitis occurs when the endodontic treatment fails to eradicate the intraradicular infection, and is mainly caused by Gram-positive bacteria and yeasts, such as *Enterococcus faecalis* and *Candida albicans*, respectively. Phenothiazines have been described as potential antimicrobials against bacteria and fungi. This study aimed to investigate the antimicrobial potential of promethazine (PMZ) and chlorpromazine (CPZ) against *E. faecalis* and *C. albicans* dual-species biofilms. The susceptibility of planktonic cells to phenothiazines, chlorhexidine (CHX) and sodium hypochlorite (NaOCl) was initially analyzed by broth microdilution. Interaction between phenothiazines and CHX was examined by chequerboard assay. The effect of NaOCl, PMZ, CPZ, CHX, PMZ + CHX, and CPZ + CHX on biofilms was investigated by susceptibility assays, biochemical and morphological analyses. Results were evaluated through one-way ANOVA and Tukey’s multiple comparison post-test. PMZ, alone or in combination with irrigants, was the most efficient phenothiazine, capable of reducing cell counts, biomass, biovolume, carbohydrate and protein contents of dual-species biofilms. Neither PMZ nor CPZ increased the antimicrobial activity of CHX. Further investigations of the properties of phenothiazines should be performed to encourage their use in endodontic clinical practice.

## 1. Introduction

Apical periodontitis is a chronic inflammatory disorder of periradicular tissues caused by microorganisms associated with endodontic infection [1]. Persistent apical periodontitis occurs when the endodontic treatment fails to eradicate the intraradicular infection [2], which is mainly caused by Gram-positive facultative anaerobes, including *Streptococcus*, *Enterococcus*, *Lactobacillus* and *Actinomyces* [3]. In addition, *Candida* species have also been isolated from infected root canals after treatment [4].

*Enterococcus faecalis* and *Candida albicans* are commensals in many niches in the human body, such as the oral cavity and the gastrointestinal mucosa [5], and are both considered endodontic pathogens associated with persistent apical periodontitis [6]. Enterococci show multi-resistance to several drugs and endodontic irrigants [7,8,9]. *C. albicans* is isolated from infected root canals in pure culture or associated with bacteria in about 5–20% of cases [10]. These two species can colonize dental tubules lateral to the canal root and resist starvation, as well as chemo-mechanical endodontic treatment [11].

The most common endodontic root canal irrigants used in conventional endodontic therapy are sodium hypochlorite (NaOCl) and digluconate chlorhexidine (CHX). NaOCl has effective antimicrobial activity against *E. faecalis* and other endodontic pathogens, even when they are organized in biofilms, and is also capable of dissolving components of the biofilm extracellular matrix and necrotic pulp tissue remnants. However, NaOCl presents high cytotoxicity and may damage the mechanical properties of dentin [12,13,14,15]. CHX is the main alternative to NaOCl, presenting greater biocompatibility and the ability to bind to dentin, exerting a prolonged antimicrobial effect [16]. However, CHX is less effective than NaOCl, especially in sessile communities, including *E. faecalis* biofilm [17]. Therefore, new endodontic therapeutic strategies, such as new irrigating substances that act synergistically with chlorhexidine, should be investigated.

Phenothiazines are a group of organic heterocyclic compounds containing nitrogen and sulfur, related to thiazine compounds, and classified as dopamine receptor agonists. The synthesis of primary phenothiazine occurs by the reaction of diphenylamine with sulfur, yielding a product that is used for the synthesis of several derivates [18]. This group is characterized by the presence of a 10H-dibenzo-[b,e]1,4 thiazine system, and has several biologic effects, such as antihistaminic, neuroleptic, antitumor, antispasmodic, anthelmintic, antibacterial, antiviral, analgesic and redox effects. Promethazine is synthesized by alkylating the phenothiazine with 1-dimethylamino-2-propyl chloride. The reaction between 3-chloro-n-phenylbenzamine and the sulfur of the phenothiazine will generate the 2-chloro-10H-phenothiazine, whose alkylation will result in the synthesis of Chlorpromazine [18]. In clinical practice, they are commonly used for the treatment of schizophrenia, nausea, methemoglobinemia, anxiety disorders and porphyria [19]. Recently, many studies have described the antimicrobial potential of the phenothiazine derivates promethazine and chlorpromazine against planktonic and sessile cells of both bacterial and fungal pathogens [20,21,22,23,24]. Rahbar et al. (2010) [25] observed an antimicrobial effect of promethazine and chlorpromazine against planktonic growth of *Enterococcus* sp. Furthermore, Galgóczy et al. (2011) and Ells et al. (2013) described the antimicrobial effect of phenothiazines against planktonic cells and biofilms of *C. albicans* [26,27]. Moreover, Castelo-Branco et al. (2013) found synergy in the antimicrobial activity between PMZ and antifungals against *C.albicans* [28].

To date, no study found in the literature was conducted to investigate the use of phenothiazines against micro-organisms related to endodontic infection in planktonic and biofilm forms of growth, nor if they could enhance the antimicrobial activity of endodontic irrigants, such as chlorhexidine.

This study aimed to investigate the antimicrobial potential of promethazine (PMZ) and chlorpromazine (CPZ) against planktonic *E. faecalis* and *C. albicans* and their dual-species biofilm. The interaction of these drugs with chlorhexidine (CHX) was also investigated.

## 2. Results and Discussion

### 2.1. Effect of Phenothiazines against Planktonic E. faecalis and C. albicans

The MICs of the compounds tested against *E. faecalis* and *C. albicans* are summarized in Table 1. MIC values for NaOCl against *E. faecalis* were higher than those against *C. albicans* (four-fold higher). MICs for CHX were the same against both microorganisms (4 μg/mL). MIC values for PMZ and CPZ were higher against *C. albicans* (3-fold, 2-fold and 1.5 higher, respectively) than *E. faecalis*. No synergistic interactions were observed for the combination of phenothiazines and CHX against *E. faecalis* and *C. albicans* (FICI: 0.5–4.0). However, two- to four-fold reductions in CHX MICs were observed against both *E. faecalis* and *C. albicans*, when it was associated with PMZ or CPZ. Moreover, PMZ and CPZ MICs suffered 8-fold and 128-fold reductions, respectively.

### 2.2. Effect of Phenothiazines on Dual-Species Biofilms

#### 2.2.1. Colony-Forming-Unit Counts and Crystal Violet Staining Assay

When compared to the control group, NaOCl significantly reduced *E. faecalis* CFU-counts and eradicated *C. albicans*, being the most efficient test solution. CHX significantly reduced *C. albicans* CFU-counts but did not affect those of *E. faecalis.* PMZ significantly reduced *E. faecalis* CFU-counts and eradicated *C. albicans*. CPZ eradicated *C. albicans*, while having no significant effect on *E. faecalis* CFU-counts. PMZ + CHX reduced both *E. faecalis* and *C. albicans* CFU-counts, while CPZ + CHX was effective only against *C. albicans* (Figure 1A,B). All the tested solutions significantly reduced the biomass of dual-species biofilm when compared to that of the growth control (Figure 1C).

#### 2.2.2. Confocal Laser Microscopy

The control group showed a dense biofilm that consisted of both cocci and yeast cells (Figure 2A). After treatment with NaOCl, there was an evident reduction in the biofilm density, displaying only dead cells (Figure 2B). Treatment with CHX resulted in some dead cocci and yeast cells, but not as remarkable as that observed after NaOCl treatment (Figure 2C). PMZ and CPZ treatment killed both yeast and cocci cells (Figure 2D,F), while the combination with CHX seemed to decrease the antibiofilm effect of phenothiazines (Figure 2E,G). All test solutions significantly decreased biofilm biovolume (Figure 2H). Biofilm thickness was significantly reduced by NaOCl, CHX, PMZ and CPZ, but not by PMZ-CHX and CPZ + CHX combinations (Figure 2I).

#### 2.2.3. Biochemical Composition

Chitin, total carbohydrates and protein contents of dual-species biofilms are shown in Figure 3. All tested solutions, except CHX and CPZ + CHX, significantly reduced the chitin content of dual-species biofilms, with the greatest reduction observed after PMZ and PMZ + CHX treatments (Figure 3A). All solutions significantly reduced the general carbohydrate content, with the greatest reduction achieved after NaOCl and PMZ treatments (Figure 3B). The protein content was significantly reduced only by NaOCl and PMZ, with greater reduction caused by the former (Figure 3C).

### 2.3. Discussion

The emergent need for new approaches to endodontic treatment encourages the search for alternative irrigant substances. The main goals of endodontic treatment are to shape and disinfect the root canals of the teeth, which is accomplished by chemical and mechanical treatments. Although there have been many technological advances in mechanical preparation devices, irrigation is still an indispensable feature, reaching root canal areas that are not accessible when using endodontic instruments [29]. The ideal irrigant for endodontic treatment must have the potential to dissolve inorganic and organic substances and present broad antimicrobial activity without being cytotoxic and damaging the dental microstructure [30].

NaOCl is the main irrigant used in endodontic practice [31], as it has an efficient antibiofilm effect, disrupts the extracellular matrix of biofilms, and induces bacterial cell death. However, NaOCl is cytotoxic, delaying the healing of periradicular tissues, and its inadvertent extrusion may cause the “NaOCl accident” [32]. Furthermore, NaOCl may alter the elasticity, flexural strength and the micro-hardness of dentine matrix [12,13,14,15,33]. NaOCl is used at a wide concentration range, varying from 0.5% to 8.25% [34], and this lack of consensus may explain the high variability in its anti-biofilm effect, as reported in the literature. Higher concentrations may result in better antimicrobial activity of NaOCl, but also in worse side effects [12]. Recent investigations recommended 2.5% NaOCl as the ideal concentration, presenting good antimicrobial activity and low toxicity [35]; thus, this concentration was used in the present study.

CHX has been used in endodontics as a final irrigant or substitute for NaOCl. When compared to NaOCl, CHX presents some advantages, such as substantivity and less cytotoxicity [16]. However, recent studies with multi-species biofilm models proved that CHX shows a weaker antimicrobial effect than NaOCl, being unable to disrupt the extracellular matrix. Furthermore, CHX does not dissolve organic or inorganic matter, therefore, it cannot be used as the only irrigating solution [36].

Following the CLSI standards, the minimal inhibitory concentration of NaOCl against *E. faecalis* and *C. albicans* found in our study was 2500 µg/mL (1:4 dilution) and 625 µg/mL (1:16), respectively. These findings are close to those of Arslan et al. (2011) [37]. In the planktonic assays, the MIC of both phenothiazines against *Candida albicans* were higher than those against *Enterococcus faecalis.* However, in the biofilm susceptibility assays, *C. albicans* were eradicated from the dual-species biofilm. These differences between assays in planktonic and dual-species biofilm growth could be explained by the fact that:In the biofilm susceptibility assay, the concentration used was 10× higher than the MICs of promethazine (1280 µg/mL) and chlorpromazine (500 µg/mL) obtained against planktonic growth;The presence of *E.faecalis* in the dual-species biofilm may modulate the susceptibility of *C. albicans* to the phenothiazines, as antagonistic interactions between these pathogens have already been suggested [38].

PMZ and CPZ have already been described as promising antibacterial and antifungal agents by inhibiting efflux pumps in mono-species biofilms [39], generating reactive oxygen species, and damaging the cell membrane of bacteria [24]. As the antimicrobial property of chlorhexidine is associated with its effect on the cell membrane [40], we speculate that this could be enhanced by the efflux pump inhibiting activity of phenothiazines. Previous studies have shown that phenothiazines increase the activity of antibiotics against planktonic bacterial pathogens and *Candida* species [25,26,41,42]. Rahbar et al. (2010) and Hendricks et al. (2005) observed an improvement of the antimicrobial activity of vancomycin and ampicillin against *Enterococcus* sp. when associated with phenothiazines. Galgóczy et al. (2013) described the increased susceptibility of *Candida* species to Amphotericin B after association with phenothiazines. Regarding the planktonic susceptibility to phenothiazines, the present results corroborate other studies evaluating the effect of PMZ and CPZ against *E. faecalis* and *C. albicans* [25,26].

PMZ and CPZ have also shown inhibitory effects against mono-species biofilms of both bacterial and fungal pathogens [23,24,27]. Evaluating the effect of phenothiazines on fungal biofilms, Ells et al. (2013) [27] described the capacity of phenothiazines to significantly reduce the biomass of *C. albicans* biofilms when compared to control (*p* < 0.05). The present study found that PMZ and CPZ affected *C. albicans* viability and, therefore, it is possible to speculate that the antibiofilm effect of both phenothiazines in dual-species biofilm may be related to their effect against *C. albicans*. Regarding the effect of phenothiazines in bacterial biofilms, Aguilar-Vega et al. (2021) [24] described that both PMZ and CPZ decreased *A. baumannii* biofilms, and CPZ was capable of inducing bacterial cell membrane damage. In the present study, both phenothiazines reduced biofilm biomass, although only PMZ reduced *E. faecalis* CFU-counts. This may be due to the different cell wall architecture or virulence traits between the two bacterial species that resulted in different CPZ susceptibility.

The biochemical analysis of dual-species biofilms after the different treatments may translate the effect of the tested substances on biofilm cells and extracellular matrix. The chitin content was significantly reduced after NaOCl, PMZ, CPZ and PMZ + CHX treatments, which is in accordance with the effect of these substances on *C. albicans* CFU. The biofilm protein content was only significantly reduced after NaOCl and PMZ treatments, with NaOCl as the most effective. Biofilm general carbohydrate content was significantly reduced after all treatments, especially after exposure to NaOCl and PMZ. Carbohydrates present in the microbial cell wall and biofilms play an important role in the host immune response against infection [43,44]. Mostly, clinical symptoms of periapical periodontitis are associated with the periodontal immune response to the biofilm colonizing the root canal system [1]. Therefore, it is feasible to speculate that after the treatment with the tested substances, the pathogens would induce a lower inflammatory response of the periodontal tissues. The capacity of NaOCl to react and disrupt amino acids and carbohydrates has already been described [45], which is associated with the formation of hypochlorous acid. Brilhante et al. (2017) [46] found that promethazine reduced the efflux of rhodamine 6G, decreased the cell size, granularity, and caused membrane damage and mitochondrial membrane depolarization in *C*. *tropicalis*. CHX had no effect on chitin content; however, the ability of CHX to reduce the general carbohydrate content of biofilms has been previously described [47], which is corroborated by our findings.

The present study has found no synergy between the antimicrobial activity of CHX and the phenothiazines PMZ and CPZ. Furthermore, CLSM and biochemical composition analysis revealed that the reduction of biofilm thickness and carbohydrate and protein content caused by PMZ and CPZ treatment was hampered after association with CHX. Subjectively, we could infer by the CLSM images that the antibiofilm effects of phenothiazines were lower than those of NaOCl, but stronger than those of CHX alone and in combination with phenothiazines. This lack of synergy may be due to the competition for the target of action. As chlorhexidine exerts its antimicrobial effect by affecting the cell membrane [40], it could be preventing the inhibition of efflux pumps, present on the cell membrane, by phenothiazines. Recently, previous studies have already described synergistic [48,49] and indifferent [50] interaction between CHX and other antimicrobials. Moreover, the induction of multidrug resistance on *E. faecalis* after exposure to sub-lethal doses of CHX has already been reported [51].

Regardless of the promising antibiofilm effect of PMZ and CPZ described in the present study, several other aspects must be considered before recommending these substances as endodontic irrigants, such as biocompatibility, effect on dentin matrix and dissolution of organic matter. Even though these drugs are clinically used for several purposes, their dental use is still poorly investigated. In the endodontic research field, Batinic et al. (2018) [52] used phenothiazine derivates as dyes in photodynamic therapy against monospecies biofilms of *E. faecalis*; however, the individual antimicrobial effect of the phenothiazine was not assessed. It is also important to emphasize that phenothiazines are mainly used to treat systemic diseases; therefore, their effect as endodontic irrigants in local administration may lead to different results. Moreover, it is feasible to speculate that the concentration of the phenothiazines used in the present study should be adjusted based on the conditions of the oral environment, which includes pH variations, osmolarity and organic substances that may affect the antimicrobial activity of the tested drugs.

Within the limitations of the present study, the cytotoxicity of the tested solutions was not investigated. It is well-known that an increase in concentration could lead to a more effective antimicrobial activity, however it also elevates the cytotoxicity of the compound [53]. In spite of the relative lower cytotoxicity of CHX, when compared to NaOCl, this cationic bisbiguanide in high concentrations could present some side effects on host cells [12]. Therefore, optimizing the antimicrobial activity of CHX without increasing its concentration seems more reasonable. This approach could be applied by combining CHX with other antimicrobial substances. However, PMZ and CPZ have the capacity to induce cell damage and release reactive oxygen species, which could contribute to their cytotoxicity [24,46]. Therefore, the authors suggest that the cytotoxicity of PMZ and CPZ on cells of the periodontal tissues (such as fibroblasts), should be investigated in further studies.

## 3. Material and Methods

### 3.1. Strains and Media

*E. faecalis* ATCC 29212 and *C. albicans* ATCC 10231 were purchased from the American Type Culture Collection (ATCC). Microorganisms were stored at −20 °C in 5% glycerol. Unfrozen microbial cells were transferred to Brain Heart Infusion broth (BHI, Kasvi, Brazil) and incubated overnight at 37 °C. Then, one loop of the culture was transferred to Mitis Salivarius agar (Sigma-Aldrich, St. Louis, MO, USA) and Sabouraud agar (Kavsi, Brazil) for *E. faecalis* and *C. albicans*, respectively. Cultures were incubated for up to 18 h at 37 °C.

### 3.2. Antimicrobial Susceptibility Assays

The antimicrobial susceptibility assays were carried out against planktonic *E. faecalis* and *C. albicans* by the broth microdilution method in 96-well U-bottomed microplates, as standardized by the documents M07-A11 [54] and M27-A3 [55], respectively. Sodium hypochlorite (NaOCl; Clororio, Brazil), chlorhexidine (CHX; Farmacia Escola UFC, Fortaleza, Brazil), promethazine (PMZ; Safoni, Avantis, Brazil) and chlorpromazine (CPZ; Safoni, Avantis, Brazil) were diluted in Mueller Hinton broth (Sigma-Aldrich, St. Louis, MO, USA) and RPMI broth (Merk, Darmstadt, Germany), for *E. faecalis* and *C. albicans*, respectively, as follows: NaOCl (19.5–10,000 µg/mL), CHX (0.062–32 µg/mL), PMZ (1–512 µg/mL), CPZ (0.25–500 µg/mL). Vancomycin (VAN; Sigma-Aldrich, St. Louis, MO, USA) and amphotericin B (AMB; Sigma-Aldrich, St. Louis, MO, USA) were also tested against *E. faecalis* and *C. albicans*, respectively, as drug controls [54,55].

Inocula were prepared from cultures previously grown on Mitis Salivarius agar (Sigma-Aldrich, St. Louis, MO, USA) for *E. faecalis* and Sabouraud dextrose agar for *C. albicans* (Kasvi, Brazil) for 24 h at 37 °C. Cell suspensions were prepared in sterile saline solution until reaching 5 × 10^5^ CFU/mL and 2.5 × 10^3^ CFU/mL for *E. faecalis* and *C. albicans*, respectively. Minimum inhibitory concentrations (MIC) were defined as the lowest concentrations capable of inhibiting 100% of microbial growth [25]. The experiments were performed at two separate moments, each with two technical replicates.

### 3.3. Chequerboard Interaction Assay between Phenothiazines and Chlorhexidine

The interaction assay was performed according to Odds (2003) [56] to evaluate the interaction between CHX and phenothiazines. Briefly, several combinations of CHX (0.156–8 µg mL^−1^) and PMZ (1–512 µg mL^−1^) or CPZ (0.25–500 µg mL^−1^) were transferred to 96-U-bottomed wells microplates and inoculated with the planktonic cells of *E. faecalis* (cell density of 5 × 10^5^ cfu mL^−1^) or *C. albicans* (cell density of 2.5 × 10^3^ cfu mL^−1^), incubated at 35–37 °C overnight. Minimum inhibitory concentrations of the drug combinations were considered the lowest concentration at which 100% inhibition of microbial growth was observed. The interaction between drugs was determined based on fractional inhibitory concentration index (FICI), being classified as synergism (FICI ≤ 0.5), antagonism (FICI > 4.0) or indifferent interaction (FICI > 0.5–4.0). The experiment was performed at two separate moments, each with two technical replicates.

### 3.4. Biofilm Formation

Biofilm formation was performed following the protocol described in a previous work of our group [38]. Colonies of each microorganism previously grown on Mitis Salivarius agar or Sabouraud agar were transferred to BHI broth or Yeast Nitrogen Base with 5% dextrose broth (YNB-D), for *E. faecalis* and *C. albicans*, respectively. The cultures were maintained for up to 18 h at 37 °C. After that, cells were harvested by centrifugation at 1591× *g* during 5 min at 4 °C, and then washed twice in sterile Phosphate-Buffered Saline (PBS). Pellets were suspended in BHI broth for each microorganism separately. Suspensions were then adjusted to 0.5 McFarland scale, corresponding to a cell density of 1.5 × 10^8^ cfu ml^−1^ for *E. faecalis* and 1.0 × 10^6^ cfu ml^−1^ to *C. albicans.* Aliquots of 100 µL of each microbial suspension (1:1 ratio) were transferred to 96-flat-bottomed-well polystyrene plates and incubated at 35 °C for 48 h in a microaerophilic atmosphere. The culture medium was renewed after 24 h by carefully removing it without damaging the biofilm on the wells and refilling them with fresh medium.

### 3.5. Biofilm Susceptibility Assays

The biofilm susceptibility assays were performed with NaOCl, CHX, PMZ, CPZ, PMZ + CHX and CPZ + CHX. The concentrations of 25 mg/mL (2.5%) NaOCl and 20 mg/mL (2%) CHX were chosen based on their well-documented use in endodontic research [29]. The concentrations for PMZ, CPZ, PMZ + CHX and CPZ + CHX were 10× MIC values for each drug against planktonic *E. faecalis* and *C. albicans* cells [57], as follows: PMZ, 1.28 mg/mL; CPZ, 0.5 mg/mL; PMZ + CHX, 0.08 mg/mL + 0.02 mg/mL; CPZ + CHX, 0.125 mg/mL + 0.020 mg/mL). Control groups were treated with sterile PBS. After 48 h of biofilm formation, the medium was gently removed, the wells were washed once with 200 µL sterile PBS, and then 200 µL of each tested substance were added. After 10 min, in order to simulate the time that endodontic irrigants remain within the root canals during the endodontic treatment [8], the solutions were removed from the wells. NaOCl groups received an additional wash with 5% sodium thiosulfate for 10 min (Dinamica, Sao Paolo, Brazil), and the other groups received PBS, in order to stop their antimicrobial activity [8]. The antibiofilm effect of each tested substance was evaluated as described below.

### 3.6. Colony Forming Units Assay

The viability of *E. faecalis* and *C. albicans* in dual-species biofilms was analyzed by counting colony-forming units (CFU). After medium removal, mature biofilms formed in 96-well plates received 200 μL of sterile saline solution and were scraped using a pipette tip. Microbial suspensions were then transferred to sterile microcentrifuge tubes and serial dilutions were made in sterile saline solution. CFU counts were performed on Mitis Salivarius agar and Sabouraud Chloramphenicol agar for *E. faecalis* and *C. albicans*, respectively. The plates were covered and sealed with Parafilm M^®^ (Merck KGaA, Darmstadt, Germany), and incubated in a microaerophilic atmosphere at 35 °C, for 24 h. Colonies were counted and CFU/mL^−1^ numbers were determined. The experiment was performed at two separate moments, each with two technical replicates.

### 3.7. Crystal Violet Assay

The biofilm biomass was analyzed using the crystal violet (CV) staining method as previously described [58]. For such, biofilms grown for 48 h in 96-flat-bottomed well plates were fixed with 200 μL of 99% methanol (Sigma-Aldrich, St. Louis, MO, USA) for 5 min. After methanol removal, the wells were dried at room temperature. Then, the biofilms were stained with 200 μL of CV stain (Sigma-Aldrich, St. Louis, MO, USA) for 20 min. After stain removal, the wells were washed twice with 200 μL of distilled water. Finally, 200 μL of acetic acid (33%) (Sigma-Aldrich, St. Louis, MO, USA) were added to each well. Absorbance was read at 590 nm using a microtiter plate reader. The experiment was performed at two separate moments, each with three technical replicates.

### 3.8. Confocal Laser Scanning Microscopy

The biofilms were formed on the surface of Thermanox™ coverslips (Thermo Fisher Scientific, Waltham, MA, USA) in 24-well plates for 48 h, with medium change after 24 h. Following 10 min of treatment with the solutions, as previously described, the biofilms were stained with LIVE/DEAD Bacterial Viability Kit (Invitrogen, Waltham, MA, USA), containing SYTO 9 and Propidium iodide. The images were acquired with a series of horizontal (x–y) optical sections, throughout the biofilm depth in a confocal microscope Nikon C2+ (Nikon, Tokyo, Japan), at a magnification of 60×. For image analysis, seven equidistant points were selected for the acquisition of biofilm three-dimensional images and the colorimetric quantification was performed with Z-slice, using the software ImageJ version 1.53t (NIH, Maryland, USA) [59]. Biomass area and average thickness were quantified using the COMSTAT plugin.

### 3.9. Biochemical Composition of Biofilms

Carbohydrate content of 48-hour-grown biofilms was analyzed by using calcofluor-white staining (Sigma-Aldrich, St. Louis, MO, USA) and Congo red and safranin staining based on previous work. The protein content was assessed by SYPRO^®^ RUBY (Thermo Fisher Scientific, Waltham, MA, USA) [38]. Briefly, after 10 min of exposure to each test-solution, biofilms were washed twice with PBS and then 1% Calcofluor-white, 0.1% Safranin and SYPRO RUBY were added to each well. The plates were maintained in the dark for 10 min and after that, the wells were washed twice with ultra-pure sterile water. Fluorescence readings for Calcofluor-white and SYPRO^®^ RUBY were performed on Cytation 3 (Biotek, Winooski, VT, USA) at 430 nm/510 nm and 465 nm/630 nm, respectively. For Congo red and safranin, readings were performed in a spectrophotometer at 490 nm and 630 nm, respectively. The experiment was performed at two separate moments, each with three technical replicates.

### 3.10. Statistical Analysis

Parametric data were analyzed using one-way analysis of variance (ANOVA) followed by Turkey’s post hoc test. *p*-values < 0.05 were considered statistically significant. Statistical analysis was performed using the software GraphPad Prism 7.0 (GraphPad Software, La Jolla, CA, USA).

## 4. Conclusions

Between the tested phenothiazines, PMZ was the most effective against *E. faecalis* and *C. albicans* dual-species biofilms. NaOCl and PMZ were capable of reducing the biomass, *E. faecalis* and *C. albicans* CFU-counts, the biovolume, the thickness and the protein–carbohydrate content of the biofilms. The association of CHX with PMZ or CPZ did not show any advantage in both planktonic and biofilm assays.

## Figures and Tables

**Figure 1 antibiotics-11-01562-f001:**
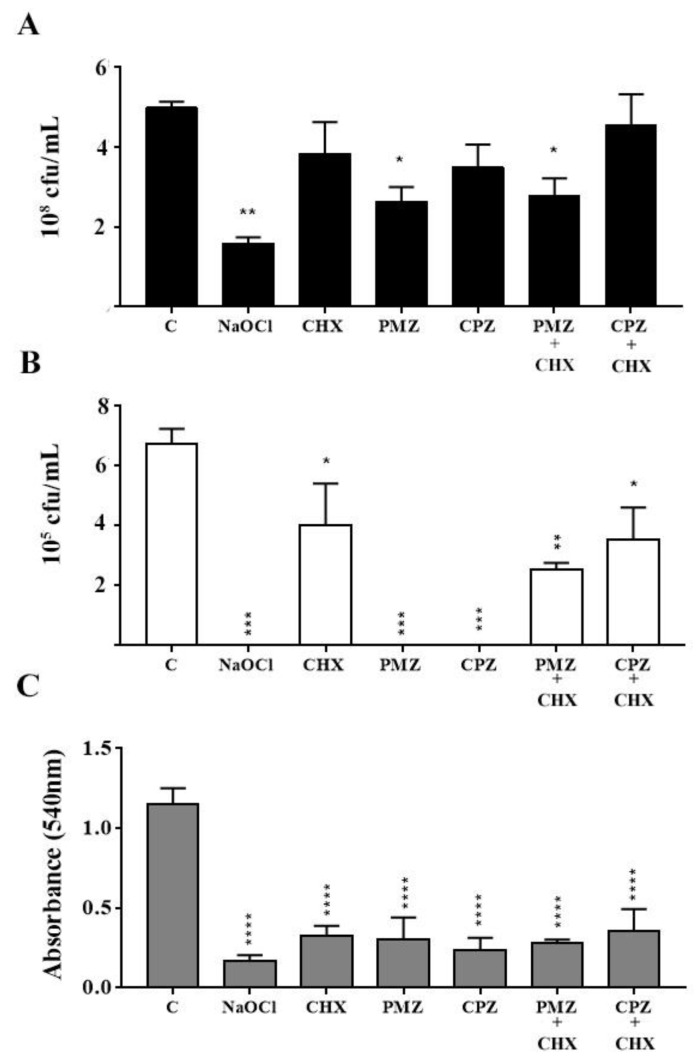
Colony-forming-unit (CFU) counts of *E. faecalis* (**A**), *C*. *albicans* (**B**) and biomass of dual-species biofilm (**C**). Mature dual-species biofilms were treated for 10 min with PBS (growth control—c) or test-solutions, as follows: NaOCl (25 mg/mL); CHX (20 mg/mL); PMZ (1.28 mg/mL); CPZ (0.5 mg/mL); PMZ + CHX (0.08 mg/mL + 0.02 mg/mL); CPZ (0.5 mg/mL) CPZ + CHX (0.125 mg/mL + 0.02 mg/mL). CFU was performed on Mitis Salivarius agar or Sabouraud agar supplemented with chloramphenicol for *E. faecalis* and *C*. *albicans*, respectively. * Represents statistically significant differences when compared to control, * represents *p* < 0.05, ** represent *p* < 0.01, *** represents *p* < 0.005 and **** *p* < 0.001. Data are expressed as mean ± SD.

**Figure 2 antibiotics-11-01562-f002:**
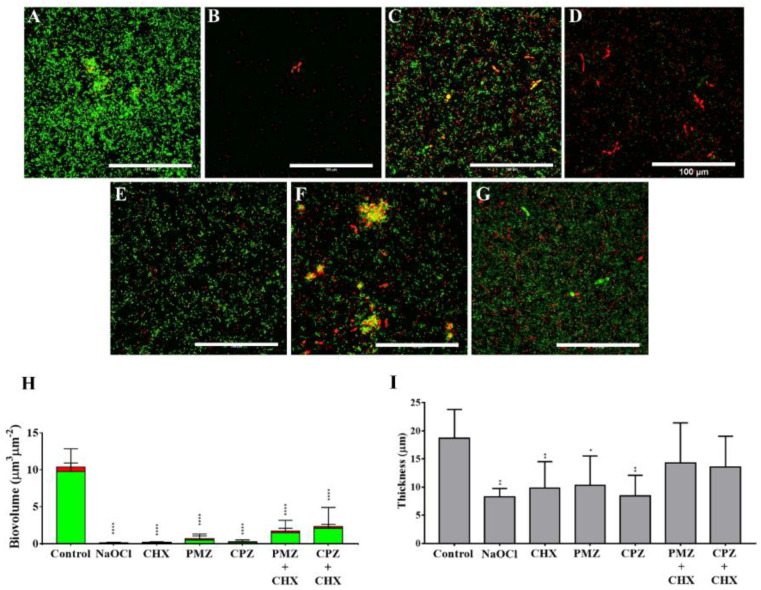
CLSM images of dual-species biofilm after treatment for 10 min with saline (**A**); 25 mg/mL NaOCl (**B**); 20 mg/mL CHX (**C**); 1.28 mg/mL PMZ (**D**); PMZ+ CHX, 0.08 mg/mL + 0.02 mg/mL (**E**); 0.5 mg/mL CPZ (**F**); CPZ + CHX, 0.125 mg/mL+ 0.02 mg/mL (**G**). Magnification 60×. Scale 100 µm. Viable cells are stained in green and dead/damaged cells are stained in red. Biovolume (**H**) and thickness (**I**) of each biofilm were quantified. * represents statistically significant differences when compared to the control. * represents *p* < 0.05, ** represent *p* < 0.01 and **** *p* < 0.001. Data are expressed as mean ± SD.

**Figure 3 antibiotics-11-01562-f003:**
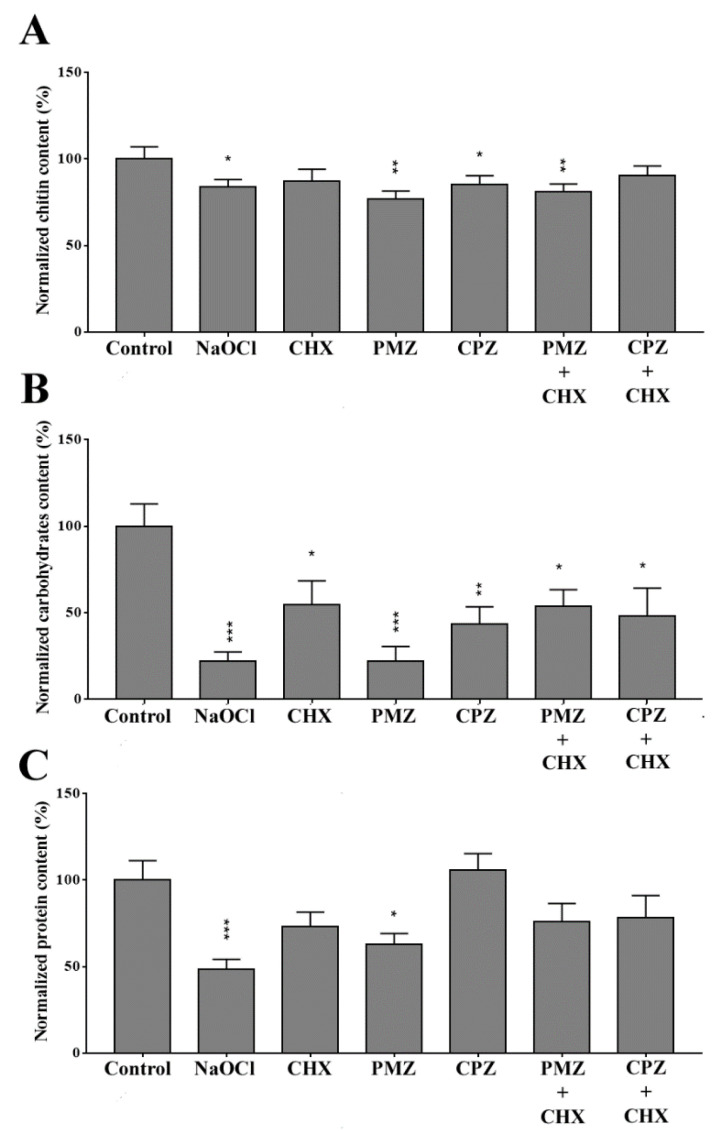
Normalized chitin (**A**), carbohydrate (**B**) and protein (**C**) content of the dual-species biofilms after treatment for 10 min with sterile saline (control); 25 mg/mL NaOCl; 20 mg/mL CHX; 1.28 mg/mL PMZ; PMZ+ CHX, 0.08 mg/mL + 0.02 mg/mL; 0.5 mg/mL CPZ; CPZ + CHX, 0.125 mg/mL + 0.02 mg/mL. * represents statistically significant differences compared to the control. * represents *p* < 0.05, ** represent *p* < 0.01, and *** *p* < 0.001. Data are expressed as mean ±SD.

**Table 1 antibiotics-11-01562-t001:** Minimum inhibitory concentration (MIC) of NaOCl, CHX, PMZ and CPZ alone and in combination against planktonic *E. faecalis* and *C. albicans*. The results are expressed in μg/mL.

Species	NaOCl	CHX	PMZ	CPZ	PMZ + CHX		CPZ + CHX	
					MIC	FICI	MIC	FICI
*E. faecalis* ATCC 29212	2500	4	64	32	8 + 1	0.562	0.25 + 1	0.507
*C. albicans* ATCC 10231	625	4	128	50	8 + 2	0.531	12.5 + 2	1

NaOCl: sodium hypochlorite, CHX: chlorhexidine; PMZ: promethazine; CPZ: chlorpromazine. FICI: Fractional inhibitory concentration index.

## Data Availability

Not applicable.

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
