# Peer review of "The Potential of Phenothiazines against Endodontic Pathogens: A Focus on Enterococcus-Candida Dual-Species Biofilm"

_antibiotics, 2022, doi:10.3390/antibiotics11111562_

Round 1
Reviewer 1 Report
The aim of this study was to investigate the antimicrobial potential of promethazine (PMZ) and chlorpromazine (CPZ) against E. faecalis and C. albicans dual-species biofilms.
The rationale for the hypothesis of a synergistic effect between phenothiazines and the endodontic irrigants (NaOCl and CHX) is not clear. The authors cited previous studies on the synergistic effects of these compounds and antibacterials/antifungals but did not explore this properly. What could be the mechanisms involved that suggest a similar synergistic effect with NaOCl or CHX?
Abstract
This statement “As expected, NaOCl was the most efficient irrigant against biofilms.” is not appropriate in the abstract.
The conclusion should be revised “Further investigations of the properties of phenothiazines should be performed to suggest their use in endodontic clinical practice.”, it is not supported by the results obtained.
Line 52 - Include more references that support for this statement “NaOCl may damage the mechanical properties of dentin”.
Line 61 – In “promethazine and chlorpromazine against planktonic and sessile cells of both bacterial and fungal pathogens [15–19].”, describe better the findings on the previous effect of these compounds specifically on E. faecalis and C. albicans.
Line 73 – What could be the importance of comparing the MIC for C. albicans and E. faecalis? in “MIC values for NaOCl against E. faecalis were higher than those against C. albicans (4-fold higher)” and “MIC values for PMZ and CPZ were higher against C. albicans (3-fold, 2-fold and 1.5 higher, respectively) than E. faecalis.”
Line 208 – Explore and describe better this statement “Previous studies have shown that phenothiazines increase the activity of antibiotics against planktonic cells of bacterial pathogens and Candida species” – What has been previous reported for E. faecalis? For Candida species, what were the antifungals tested in these studies? What are the correlation of these previous studies with this study? What are the rationale for expecting a synergistic effect between phenothiazines and the endodontic irrigants tested?
Line 233 – The statement “It is possible to speculate that the reduction of structural carbohydrates within the extracellular matrix of dual-species biofilms contributes to the large biomass reduction observed.” is poorly supported by evidences and should be better discussed.
Line 235 – In “This result is of great importance, seeing that the carbohydrates present in the yeast cell wall are also related to host immunogenic response against fungal infection.”. This paragraph is confusing. Consider revising. Clarify this statement considering the impact of host immunogenic response in the root canal.
The discussion section should include a discussion on the clinical application of the findings. For instance, what could be the clinical application of this finding “PMZ was more effective than CPZ against E. faecalis and C. albicans dual-species biofilms”?
Conclusions: If PMZ or CPZ had no synergistic effect both with CHX and NaOCl, why does further research evaluating properties of clinical interest needed?
English review is needed, i.e., chequerboard assay
Minor comment
Line 37 – Italics are missing.
Reviewer 2 Report
The manuscript describes the antimicrobial effects of promethazine and chlorpromazine as well as chlorhexidineon on Enterococcus faecalis and Candida albicans, and their biofilms. The topics of the manuscript seem suitable for publication in Antibiotics.
[Suggestions]
>> Table 1 and Figure 1(B):
Since the referee feels that the data of minimum inhibitory concentration of promethazine and chlorpromazine (128 and 50 µg/mL, respectively; in Table 1) and the bar graph of Figure 1(B) seem a little bit inconsistent, is it possible for the authors to describe the reasons of the situations?
>> Line 130-132 and Figure 2(I):
>> The thickness was significantly reduced by NaOCl, CHX, PMZ and CPZ, but not by phenothiazine-chlorhexidine combinations (fig. 2I).
The referee wonders whether it is possible for the authors to mention "chlorpromazine-chlorhexidine" combinations.
>> L. 159-161 and Figure 3(A):
>> "All tested solutions, except CHX and CPZ+CHX, significantly reduced the chitin content of dual-species biofilms, with greatest reduction observed after PMZ and PMZ + CHX treatments."
Is it possible for the authors to modify the Figure 3(A) to emphasize the "greatest reduction after PMZ and PMZ + CHX treatments", but "NOT after CHX and CPZ + CHX treatments".
[Typographical errors]
>>L. 159, 161 and 164:
"Figure 4" should read "Figure 3".
Reviewer 3 Report
The manuscript antibiotics-1986004 "The potential of phenothiazines against endodontic pathogens: a focus on Enterococcus-Candida dual-species biofilm" by Nicole de Mello Fiallos et al. describes the study of antibacterial activity of phenothiazine derivatives, i.e. promethazine (PMZ) and chlorpromazine (CPZ), and their combination with chlorhexidine (CHX) against E. faecalis and C. albicans dual-species biofilms.
The authors obtained interesting biological results, so I think this paper will be of interest to the readers of the Antibiotics journal. However, I have some questions and comments.
1) The antibacterial activity of the studied compounds has been known for a long time. The authors need to write more clearly what is the novelty of this manuscript.
2) What was the proposed mechanism for improving the activity of the studied combination of antibiotics? How can the authors confirm this? Why were these phenothiazine derivatives chosen? Why didn't the CHX combination idea work?
3) The manuscript contains a weak discussion of the obtained results. What correlations and recommendations can the authors make? How is the “Biochemical composition” part related to the previous ones? The conclusions also poorly correlate well with previous chapters.
4) I recommend the authors to strengthen the Introduction part about synthesis and applications of phenothiazine derivatives. Recent review articles of 2022 on this topic should be added. For example, 10.3390/molecules27010276, 10.1039/D2TC02085H, 10.1016/j.dyepig.2022.110806.
5) The authors wrote about the high toxicity of some components, i.e. NaOCl and CHX. What about the toxicity of the obtained combination of the studied compounds?
6) I recommend comparing the results obtained by the authors with previous results of the CHX and antibiotics combination obtained by other scientific groups.
7) Minor changes:
- The manuscript has different font sizes
- Lines 156-165. Please check the numbering of the Figures (Figure 4 is not in the manuscript).
-Please check superscripts and subscripts, i.e. mg ml−1 should be mg×ml−1 or mg/ml.
Round 2
Reviewer 1 Report
The manuscript was revised by the authors. Two points should be still addressed by the authors:
1. In discussion, line 289, "Even though these drugs have already been used in clinics to several purposes, their dental approach is still poorly investigated": please include a deeper discussion, considering that the concentrations and mode of administration are diverse. The current use in clinics is for systemic diseases, as described in line 69 and in the present study the authors suggested a topical use (endodontic irrigant).
2. In discussion, clarify "What could be the main advantages of PMZ when compared to NaOCl"? Is there any previous information on the toxicity of PMZ when applied topically? If so, include this information.
Reviewer 2 Report
>> Table 1 and Figure 1(B):
>> Please let us know if the following text should be added to the manuscript:
Yes, it should be included in the manuscript.
>> Line 130-132 and Figure 2(I):
>> Please, find the review in lines 153-155 (highlighted and tracked)
In lines 147-149???
>> to modify the Figure 3(A) to emphasize
The bar graph of the Figure 3A seems a little bit similar, although the referee has recognized the meanings the authors would like to say .....
>> "Figure 4" should read "Figure 3".
>> Please, find the review in lines 184,186,187 (highlighted and tracked)
In lines 175,177,179-180??? And, in line 175, it still should be Table 3 (instead of Table 4).
Reviewer 3 Report
I thank the authors for answering my questions and improving the manuscript.
